# Contrastive Imitation Learning for Language-guided Multi-Task Robotic Manipulation

**Teli Ma**[1]    **Jiaming Zhou**[1]    **Zifan Wang**[1]    **Ronghe Qiu**[1]    **Junwei Liang**[1,2*]

[1]AI Thrust, The Hong Kong University of Science and Technology (Guangzhou)
[2]CSE Department, The Hong Kong University of Science and Technology

[Σ-agent.com](Σ-agent.com)

**Abstract:** Developing robots capable of executing various manipulation tasks, guided by natural language instructions and visual observations of intricate real-world environments, remains a significant challenge in robotics. Such robot agents need to understand linguistic commands and distinguish between the requirements of different tasks. In this work, we present Σ-agent , an end-to-end imitation learning agent for multi-task robotic manipulation. Σ-agent incorporates contrastive Imitation Learning (contrastive IL) modules to strengthen vision-language and current-future representations. An effective and efficient multi-view querying Transformer (MVQ-Former) for aggregating representative semantic information is introduced. Σ-agent shows substantial improvement over state-of-the-art methods under diverse settings in 18 RLBench tasks, surpassing RVT [1] by an average of $5.2\%$ and $5.9\%$ in 10 and 100 demonstration training, respectively. Σ-agent also achieves $62\%$ success rate with a single policy in 5 real-world manipulation tasks.

**Keywords:** Contrastive Imitation Learning, Multi-task learning, Robotic Manipulation

## 1   Introduction

One of the ultimate goals of robotic manipulation learning is to enable robots to perform a variety of tasks based on instructions given by humans in natural language. This requires robots to understand and distinguish minor variations in linguistic commands and visual cues. However, training robots is difficult due to the limited rewards available in simulated environments and the lack of extensive real-world data. Imitation learning is an effective off-policy method that avoids complex reward design and low-efficient agent-environment interactions [2, 3, 4, 1]. In this paper, we focus on imitation learning for 3D object manipulation.

Previous works have mainly concentrated on improving the perception ability of robotic agents, but ignoring the development of discriminating different instructions and related tasks. A portion of these studies has been directed towards enhancing the transferability from 2D pre-trained visual representation to the real-world [2, 7, 8, 9]. Nonetheless, to maintain geometric details for both simulated and real-world environments, 3D vision learning dominates in instruction-guided manipulation [4, 3, 10, 1, 11, 12, 13, 14]. For instance, C2FARM [4] leveraged 3D ConvNets to aggregate visual representations based on pre-constructed voxel space. PerAct [3] constructed voxelized observations and discrete action space on RGB-D images, utilizing Percevier [15] Transformer to encode features. Moreover, PolarNet [11] directly encoded the point cloud features reconstructed from RGB-D to predict actions. However, these works do not address how to train visual representations to align with linguistic features and differentiate between multiple tasks.

Previous methods [2, 3, 16, 11, 1, 12, 13, 14] can be summarized as training the agent to learn a parameterized policy $\pi_\theta$ that imitates the target policy $\pi^+$ based on data collected using the target

---

*Corresponding author.

8th Conference on Robot Learning (CoRL 2024), Munich, Germany.

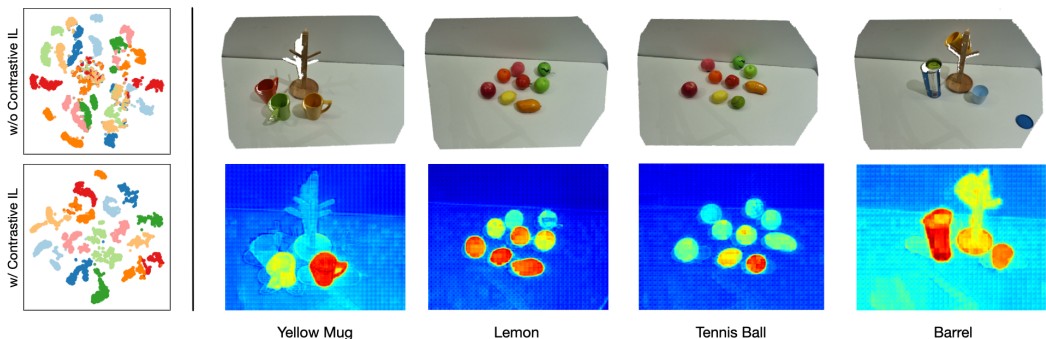

Figure 1: **Left:** t-SNE [5] visualization of multi-task representation learning without/with contrastive IL, and learning with contrastive IL shows a much more obvious separation of features belonging to different tasks. **Right:** Visualize the interested regions with Grad-CAM [6], which shows accurate object-level understanding.

policy via behavior cloning (BC) loss. These approaches typically involve mapping visual representations $\phi$, linguistic representations $\psi$, and vision-language interactions $\delta$ to low-level end-effector actions. To tackle the aforementioned misalignment problem, we introduce an end-to-end contrastive Imitation Learning (contrastive IL) strategy to the original language-conditioned policy learning, inspired by contrastive Reinforcement Learning (contrastive RL) methods [17, 18, 19, 20]. The contrastive IL framework strengthens the training process by establishing: 1) a relationship between *language instructions* and *corresponding visual inputs*, which helps differentiate multi-task representations, and 2) a link between *current* and *future states*, which enhances the recognition of movement trajectories across different tasks. This contrastive loss provides additional supervision to the feature extraction $\phi$ and interaction $\delta$ components (as shown in Fig. 2 (b)), complementing the standard BC loss.

Based on the contrastive IL, we present an end-to-end trained language-conditioned multi-task agent to complete 6-DoF manipulation, dubbed as *contraStive Imitation learninG for Multi-tAsk manipulation* agent (SIGMA-agent, abbreviated as Σ-agent ). Σ-agent follows the state-of-the-art baseline model, RVT [1], and leverages the re-rendered virtual images from RGB-D reconstruction to explicitly represent visual information. We propose a Multi-View Querying Transformer (MVQ-Former) [21, 22, 23] to minimize the number of tokens for efficient contrastive IL. The Σ-agent framework offers guidance on how to incorporate contrastive learning into existing imitation learning methods, while the inference process stays the same.

Experiments on RLBench [24] and real-world tasks demonstrate the effectiveness of the Σ-agent . The results on RLBench show Σ-agent significantly outperforms previous agents in both the 10 demonstrations (by +5.2% on average) and the 100 demonstrations (by +5.9% on average) training under the setting of one policy for 18 tasks with 249 variations. Also, we integrate the contrastive IL module into existing methods (PolarNet [11], RVT [1]) and experiment Σ-agent on another simulation environment (Ravens [25]). The significant improvements show the general applicability of our approach across various models and environments. Σ-agent also achieves 62% multi-task success rate on average with a single policy over 5 tasks in real-world robot experiments.

## 2 Related Work

### 2.1 Language-conditioned Robotic Manipulation

Language-conditioned robotic manipulation has emerged as a pivotal research branch in the robotics domain due to its extensive applicability in human-robot interaction [26, 2, 3, 1, 11, 16, 12, 13, 14, 27, 28, 29, 30]. Many previous studies have delved into vision-based representations and strategies for vision-language interaction in policy learning. For example, RT-1 [31] encodes multi-modal tokens via a pre-trained FiLM EfficientNet model and feeds them into the Transformer for multi-

modal information aggregation. The later version RT-2 [32] leverages the auto-regressive generative capacity of LLMs to project visual tokens into linguistic space, and uses LLMs to generate the actions directly. Diverse benchmarks have been curated to benchmark the language-conditioned manipulation [24, 25, 33, 34, 35, 36]. In this paper, we mainly focus on RLBench [24], which provides hundreds of challenging tasks and diverse variants covering object poses, shapes, colors, sizes, and categories to evaluate agents based on RGB-D cameras.

Numerous efforts have been made in this challenging benchmark. C2FARM [4], PerAct [3] and GN-Factor [14] harness the 3D voxel representation for policy learning. C2FARM [4] detects actions at two levels of voxelization in a coarse-to-fine manner. PerAct utilizes the Perceiver network [15] to encode 3D voxel features to predict the next keyframe positions with lower voxel resolution than C2FARM [4]. To refine the 3D scene geometry understanding, GNFactor [14] incorporates a generalized neural feature fields module to distill 2D semantic features into NeRFs [37] based on PerAct [3]. Besides the voxelized features, policy learning based on 3D point cloud representation has gained significant attention such as PolarNet [11], Act3D [12] and ChainedDiffuser [13]. PolarNet trains agents on 3D point clouds constructed from RGB-D and adopts pre-trained PointNext [38] to extract point-wise features. Both Act3D [12] and ChainedDiffuser [13] employ a coarse-to-fine sampling strategy to select 3D points in space and feature them with relative spatial attention, while ChainedDiffuser [13] synthesize end-effector trajectories with a diffusion model. Our work follows RVT [1], which re-renders virtual view images from reconstructed 3D point clouds and processes the images using a Transformer network.

## 2.2 Contrastive Learning in RL

A large volume of prior work combines representation learning objective with RL objective [39, 40, 41, 42, 43, 44]. Contrastive learning has gained significant attention among these representation learning methods [40, 41, 45, 46, 43, 47]. Recently, the paradigm of unifying the representation learning and reinforcement learning objective has emerged as a research hotspot in the field of RL [19, 45, 18, 48, 49, 50, 51]. For instance, C-learning [19] regards goal-conditioned RL as estimating the probability density over future states, learning a classifier to distinguish the positive future state from the random states. Eysenbach et al. [17] demonstrate that contrastive representation learning can be used as value function estimation, connecting the learned representations with reward maximization. Zheng et al. [18] propose to stabilize contrastive RL in offline goal-reaching tasks, analyzing the intrinsic mechanism of contrastive RL deeply to explore ingredients for stabilizing offline policy learning. Note that the contrastive RL methods above mainly focus on reinforcement learning with reward updating. One of the most similar works to ours is GRIF [52], which learns linguistic representations that are aligned with the collected transitions in the trajectory via contrastive learning. However, our contrastive IL is different from GRIF [52] in three aspects. First, contrastive IL is an end-to-end training paradigm, while the GRIF [52] decouples the contrastive representation pre-training and policy learning into two phases. Second, we target the 3D multi-task setting in this paper, whereas GRIF [52] utilizes RGB images and single policy training. Lastly, GRIF [52] performs contrastive learning on (state, goal) pairs with language instructions. For contrastive IL, we perform contrastive learning to refine both the feature extraction and vision-language feature interaction. More related work is present in Appendix D.

## 3 Method

Fig. 2 provides an overview of the Contrastive IL design in Σ-agent (Fig. 2 (a)) and how to apply to the previous language-guided imitation learning paradigm (Fig. 2 (b)). In this section, we introduce the components of Σ-agent . Additional details about Σ-agent are provided in Appendix C.

### 3.1 Preliminaries

We assume a language-conditioned Markov decision process (MDP) defined by states $s_t \in \mathcal{S}$, actions $a_t \in \mathcal{A}$ and language instructions $l \in \mathcal{L}$. $\mathcal{S}, \mathcal{A}$ are the state and action spaces, and $\mathcal{L}$

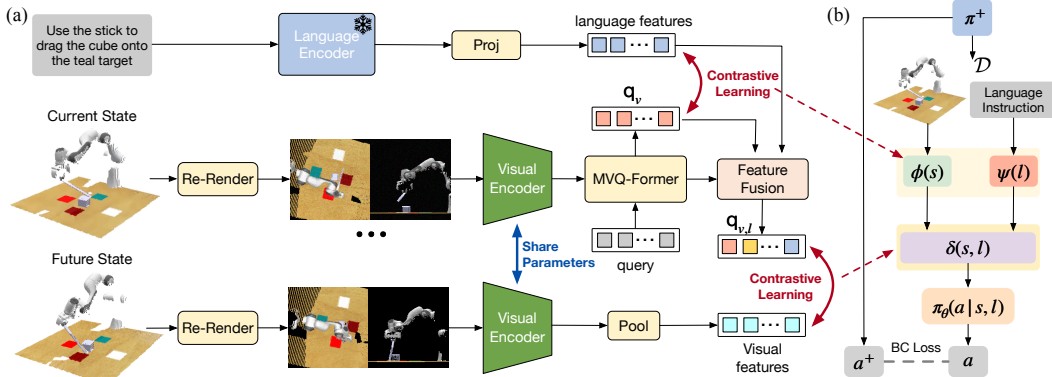

Figure 2: (a) **The pipeline of Contrastive IL in $\Sigma$-agent**. (b) **The overview of imitation learning** for language-conditioned multi-task manipulation, where representation $\phi, \psi, \delta$ and policy network $\theta$ are learned for policy $\pi_\theta$ to mimic target policy $\pi^+$. **Red Dotted Line:** The contrastive IL aim to refine visual representation $\phi$ (visual encoder) and joint vision language representation $\delta$ (MVQ-Former and Feature Fusion).

represents the set of language instructions. The goal is to learn a policy $\pi : \mathcal{S} \times \mathcal{L} \to \mathcal{A}$ to maximize the expected rewards of predicted actions. Following previous work [3], we utilize behavior cloning to maximize the Q-function without specific rewards. Therefore, the objective of the policy learning can be formulated as:

$$\theta = \arg\max_\theta \mathbb{E}_{(s_t, a_t) \sim \mathcal{D}} \log \pi_\theta(a_t | s_t, l), \tag{1}$$

where $\theta$ means parameters of the policy network and $\mathcal{D}$ represents the transitions from demonstrations collected for behavior cloning. Note that $\mathcal{D}$ is sampled from the expert policy $\pi^+$, and we train the $\theta$ to drive $\pi_\theta$ to mimic the $\pi^+$. In this paper, the state $s_t$ includes aligned RGB and depth images from the front, left shoulder, right shoulder, and wrist position. We follow RVT [1] by adopting re-rendered virtual images from the RGB-D inputs to feed into the model. The action space $\mathcal{A}$ is composed of translation in Cartesian coordinates $a_t^{trans} \in \mathbb{R}^3$, rotation in quaternion $a_t^{rot} \in \mathbb{R}^4$, gripper open $a_t^o \in \{0, 1\}$ and collision state $a_t^c \in \{0, 1\}$.

### 3.2 Visual and Language Encoder

We acquire re-rendered virtual images following RVT [1] from 5 cubic viewpoints: the front, left, right, behind and top. Each view image comprises RGB, depth, and $(x, y, z)$ coordinates channels. The visual encoder comprises a patch embedding layer and a two-layer self-attention Transformer. We split the images into $20 \times 20$ patches and leverage an MLP layer to project the embeddings of patch tokens for self-attention. For the self-attention Transformer, each patch token only attends to other tokens within the same virtual view image, which aims to aggregate the information from the same view. The visual encoder is trained from scratch with normalized initialization.

For the language encoder, we follow previous works [3, 16, 11, 1, 12] and use pre-trained language encoder from CLIP [53] for fair comparison. The language encoder remains frozen during training. The language token embeddings from the encoder are then projected by a trainable MLP for the cross-attention with visual tokens.

### 3.3 Multi-View Query Transformer (MVQ-Former)

With the extracted visual features from the visual encoder, we follow the query Transformer [23, 53, 21, 22] to pre-define a set of learnable queries. These queries are utilized by the contrastive IL module, minimizing the computational complexity by reducing the number of original visual tokens. The number of learnable queries is set to 5, one for each virtual view to aggregate the intra-view information. The MVQ-Former in Fig. 2 consists of two cross-attention layers, where the query tokens co-attend to the extracted visual features. We denote the queries produced by MVQ-Former as $\mathbf{q}_v$.

The query tokens, visual tokens and language tokens are then concatenated together and fed into 4 self-attention layers for feature fusion. During this process, the queries interact sufficiently with both the visual and language features, resulting in queries at this stage named $\mathbf{q}_{v,l}$. The $\mathbf{q}_v$ and $\mathbf{q}_{v,l}$ are utilized for contrastive IL as described in the following section. Finally, the context features (denote as $\mathbf{v}$) after the self-attention layers are input to the decoder for action prediction.

## 3.4 Contrastive Imitation Learning

For multi-task agents conditioned by language instructions, we propose the key question: *What are effective task representations for accurate, fine-grained controls?* Two factors matter. First, aligning task and language representations is essential for the agent to comprehend correlations and differentiate between tasks. Second, the agent needs discriminative features to accurately perceive the current state.

**Learning Objective.** Inspired by contrastive RL [19, 17, 20, 18], we propose to integrate the contrastive representation learning into current imitation learning framework to train the policy in an end-to-end manner. The contrastive IL comprises state-language ($s \leftrightarrow l$) and (state, language)-future ($s, l \leftrightarrow g$) contrastive learning to refine both the feature extraction and interaction as shown in Fig. 2.

For the $s \leftrightarrow l$, we define the similarity function as $f_{\phi,\psi} = \exp\left(\phi(s_t)^\top \psi(l)/\tau\right)$ ($\tau$: temperature parameter), where $\phi$ and $\psi$ are the state and language instruction representations, respectively. The contrastive IL objective between states and language instructions is:

$$
\begin{aligned}
\mathcal{L}_{s\leftrightarrow l}(s_t, l) =& \frac{1}{N} \sum_N \log \frac{f_{\phi,\psi}(s_t^+, l^+)}{f_{\phi,\psi}(s_t^+, l^+) + \sum_{l^- \in L^-} f_{\phi,\psi}(s_t^+, l^-)} \\
&+ \frac{1}{N} \sum_N \log \frac{f_{\psi,\phi}(l^+, s_t^+)}{f_{\psi,\phi}(l^+, s_t^+) + \sum_{s_t^- \in S^-} f_{\psi,\phi}(l^+, s_t^-)},
\end{aligned}
\tag{2}
$$

where $N$ is the batch size in training, and $s_t^+, l^+$ denotes corresponding positive states and instructions. $L^-$ and $S^-$ stand for all negative language and state samples in the current batch. Essentially this enforces the network to learn more discriminative representations by aligning state and language pairs in the joint embedding space.

For the $(s, l \leftrightarrow g)$, we randomly sample future states $g$ from $\mathcal{D}$, and applying network $\varphi$ to encode. $\delta$ encodes the joint representation of $(s_t, l)$. The negative samples are from the other tasks in the same batch. Similar to Eqn. 2, we formulate the contrastive training loss between the future and state, language pairs as:

$$
\begin{aligned}
\mathcal{L}_{(s,l)\leftrightarrow g}(s_t, l, g) =& \frac{1}{N} \sum_N \log \frac{f_{\delta,\varphi}((s_t^+, l^+), g^+)}{f_{\delta,\varphi}((s_t^+, l^+), g^+) + \sum_{g^- \in G^-} f_{\delta,\varphi}((s_t^+, l^+), g^-)} \\
&+ \frac{1}{N} \sum_N \log \frac{f_{\varphi,\delta}(g^+, (s_t^+, l^+))}{f_{\varphi,\delta}(g^+, (s_t^+, l^+)) + \sum_{(s_t^-, l^-) \in \Omega^-} f_{\varphi,\delta}(g^+, (s_t^-, l^-))}.
\end{aligned}
\tag{3}
$$

The $\mathcal{L}_{(s,l)\leftrightarrow g}$ aims to refine the representation of vision-language interaction $\delta$, which is crucial for the vision-language understanding of agents. $G^-$ and $\Omega^-$ signify negative space of future states and $(s_t, l)$ in the current batch. Based on the $\mathcal{L}_{s\leftrightarrow l}$ and $\mathcal{L}_{(s,l)\leftrightarrow g}$, we re-formulate the learning objective in Eqn. 1 as:

$$
\begin{aligned}
\max_\theta \quad & \mathbb{E}_{\substack{(s_t, a_t, l, g) \sim \mathcal{D}, \\ g^+ \sim \mathcal{P}_{\pi^+}(\cdot | s_t, l)}} [\lambda \log \pi_\theta(a_t | s_t, l) \\
& + (1 - \lambda) \underbrace{[\mathcal{L}_{s\leftrightarrow l}(s_t, l) + \mathcal{L}_{(s,l)\leftrightarrow g}(s_t, l, g)]}_{\mathcal{L}_{CL}}],
\end{aligned}
\tag{4}
$$

where $\lambda$ is the coefficient. The positive future state $g^+$ conforms to the transition probability $\mathcal{P}_{\pi^+}$ of the target policy $\pi^+$. More discussions can be found in Appendix C.2.

**Module Details.** For the contrastive training between language and state observations, we choose the last token (i.e., $[EOS]$) as the feature representation of the whole text following CLIP [53], which

Table 1: Multi-task performance trained with 100 episodes and evaluated using 25 episodes per task on RLBench (5 times average). The input resolution is $128 \times 128$. In `Train Time`, the $d$ represents days and $h$ represents hours. All other results are success rates, measured in percentage (%).

| | Avg Succ. | Train Time | open drawer | slide block | sweep to dustpan | meat off grill | turn tap | put in drawer | close jar | drag stick |
|---|---|---|---|---|---|---|---|---|---|---|
| I-BC(CNN) [2] | 1.3 | - | 4.0 | 0 | 0 | 0 | 8.0 | 8.0 | 0 | 0 |
| I-BC(ViT) [2] | 0.9 | - | 0 | 0 | 0 | 0 | 16.0 | 0 | 0 | 0 |
| C2FARM [4] | 16.0 | - | 20.0 | 16.0 | 0 | 20.0 | 68.0 | 4.0 | 24.0 | 24.0 |
| Hiveformer [16] | 45.3 | - | - | - | - | - | - | - | - | - |
| PolarNet [11] | 44.7 | ~20h | 81.3 | 53.3 | 52.0 | 92.0 | 78.7 | 28.0 | 37.3 | 92.0 |
| PerAct [3] | 49.4 | ~16d | **88.0**±5.7 | 74.0±13.0 | 52.0±0.0 | 70.4±2.0 | 88.0±4.4 | 51.2±4.7 | 55.2±4.7 | 89.6±4.1 |
| RVT [1] | 62.9 | ~23h | 71.2±6.9 | **81.6**±5.4 | 72.0±0.0 | 88.0±2.5 | 93.6±4.1 | **88.0**±5.7 | 52.0±2.5 | 99.2±1.6 |
| Σ-agent | **68.8** | ~22h | 76.8±3.8 | 74.4±4.5 | **80.8**±1.3 | **97.6**±1.9 | **95.2**±1.3 | 70.4±3.8 | **78.4**±2.9 | **100.0**±0.0 |

| | stack blocks | screw bulb | put in safe | place wine | put in cupboard | sort shape | push buttons | insert peg | stack cups | place cups |
|---|---|---|---|---|---|---|---|---|---|---|
| I-BC(CNN) [2] | 0 | 0 | 4.0 | 0 | 0 | 0 | 0 | 0 | 0 | 0 |
| I-BC(ViT) [2] | 0 | 0 | 0 | 0 | 0 | 0 | 0 | 0 | 0 | 0 |
| C2FARM [4] | 0 | 8.0 | 12.0 | 8.0 | 0 | 8.0 | 72.0 | 4.0 | 0 | 0 |
| HiveFormer [16] | - | - | - | - | - | - | - | - | - | - |
| PolarNet [11] | 1.3 | 41.3 | 84.0 | 41.3 | 12.0 | 8.0 | 96.0 | 1.3 | 5.3 | 0 |
| PerAct [3] | 26.4±3.2 | 17.6±2.0 | 84.0±3.6 | 44.8±7.8 | 28.0±4.4 | 16.8±4.7 | 92.8±3.0 | 5.6±4.1 | 2.4±2.0 | 4.0±2.5 |
| RVT [1] | 28.8±3.9 | 48.0±5.7 | 91.2±3.0 | **91.0**±5.2 | 49.6±3.2 | **36.0**±2.5 | 100±0.0 | 11.2±3.0 | 26.4±8.2 | **4.0**±2.5 |
| Σ-agent | **51.2**±5.4 | **73.2**±2.2 | **98.4**±1.9 | 90.4±3.5 | **66.4**±4.5 | **36.0**±3.2 | **100.0**±0.0 | **15.2**±2.9 | **33.6**±6.7 | 0.8±1.3 |

is then linearly projected into the multi-modal embedding space. The queries $\mathbf{q}_v$ are projected along the token dimension to aggregate the representative visual features as shown in Fig. 2 (a). Then, the visual token and text token are trained to align in the joint embedding space based on Eqn. 2.

We utilize the next state in the trajectory as the future state. The visual encoder of the future state shares the same backbone parameters with that of the current state as shown in Fig. 2. Hence, the $\varphi$ in Eqn. 3 equals to $\phi$ in Eqn. 2. The queries $\mathbf{q}_{v,l}$, which include both the visual features of the current state and the language features, are projected to perform contrastive training with average-pooled future-state features. This process conforms to Eqn. 3. Note that the contrastive IL module is designed to enhance representations during training but is disabled during the inference process.

## 4 Experiments

### 4.1 Simulation Experiments

**Simulation Setup.** RLBench is a robot manipulation benchmark built on CoppelaSim [54] and PyRep [55]. We follow the protocols of PerAct [3] and RVT [1] to test the model on 18 tasks in RLBench [24]. These tasks, which include picking and placing, bulbs screwing, and drawer opening *etc.*, are all performed by controlling a Franka Panda robot with a parallel gripper. Details of the 18 tasks and their variations are provided in Appendix A.1. The input RGB-D observations are obtained from four RGB-D cameras at the front, left shoulder, right shoulder, and wrist positions. The input resolution is $128 \times 128$ for experiments on RLBench unless otherwise specified.

**Implementation Details.** We follow RVT [1] to perform training on cube-viewed re-rendered images from 3D point clouds. We evaluate Σ-agent with 10 and 100 demonstrations per task for training. Following previous work [24, 56, 3, 1], we perform behavior cloning on the replay buffer of extracted keyframes rather than all frames from episodes. Similar to PerAct [3],

Table 2: RLBench results evaluated on 100 episodes per task with $256 \times 256$ input resolution.

| Method | Avg Succ. | Train Time |
|---|---|---|
| Act3D [12] | 65.1 | ~5.5d |
| ChainedDiffuser [13] | 66.1 | ~4.5d |
| Σ-agent | **68.4** | **~22h** |

we adopt translation and rotation data augmentations during training, perturbing the point clouds randomly in the range of $\pm 0.125m$ for translation and rotating the point clouds along the $z$-axis within $\pm 45°$. For the training scheme, we train Σ-agent for 25K steps with a batch size of 96 and an initial learning rate of $9.6 \times 10^{-4}$. The LAMB [57] optimizer is applied, with cosine learning rate decay and 2K warming steps. The training is conducted on $8 \times$NVIDIA A6000 GPUs for around 22 hours. Σ-agent is evaluated on all 18 tasks with variants. The initial observations are given to the agent, and the agent explores to reach the final state by the observation-action loop. The agent scores

Table 3: Multi-task performance of integrating contrastive IL into baselines. Σ-PolarNet and Σ-RVT represents PolarNet [11] and RVT [1] model trained with proposed contrastive IL module.

| Methods | Avg Succ. | Model Param.(M) | open drawer | slide block | sweep to dustpan | meat off grill | turn tap | put in drawer | close jar | drag stick |
|---|---|---|---|---|---|---|---|---|---|---|
| PolarNet [11] | 44.7 | 14.1 | 81.3 | 53.3 | 52.0 | 92.0 | 78.7 | 28.0 | 37.3 | 92.0 |
| Σ-PolarNet | 47.5 | 15.3 | 74.7 | 59.3 | 54.7 | 94.7 | 81.3 | 40.0 | 37.3 | 93.3 |
| improvement | +2.8 | - | -6.6 | +6.0 | +2.7 | +2.7 | +2.6 | +12.0 | +0 | +1.3 |
| RVT [1] | 62.9 | 36.4 | 71.2 | 81.6 | 72.0 | 88.0 | 93.6 | 88.0 | 52.0 | 99.2 |
| Σ-RVT | 64.7 | 38.2 | 82.6 | 81.3 | 53.3 | 96.0 | 94.7 | 70.7 | 65.3 | 100.0 |
| improvement | +1.8 | - | +11.4 | -0.3 | -18.7 | +8.0 | +1.1 | -17.3 | +13.3 | +0.8 |

| | stack blocks | screw bulb | put in safe | place wine | put in cupboard | sort shape | push buttons | insert peg | stack cups | place cups |
|---|---|---|---|---|---|---|---|---|---|---|
| PolarNet [11] | 1.3 | 41.3 | 84.0 | 41.3 | 12.0 | 8.0 | 96.0 | 1.3 | 5.3 | 0.0 |
| Σ-PolarNet | 9.3 | 37.3 | 85.3 | 54.6 | 16.0 | 6.7 | 100.0 | 4.0 | 4.0 | 4.0 |
| improvement | +8.0 | -4.0 | +1.3 | +13.3 | +4.0 | -1.3 | +4.0 | +2.7 | -1.3 | +4.0 |
| RVT [1] | 28.8 | 48.0 | 91.2 | 91.0 | 49.6 | 36.0 | 100.0 | 11.2 | 26.4 | 4.0 |
| Σ-RVT | 42.7 | 52.0 | 92.0 | 93.3 | 69.3 | 33.3 | 100.0 | 21.3 | 12.0 | 4.0 |
| improvement | +13.9 | +4.0 | +0.8 | +2.3 | +19.7 | -2.7 | +0 | +10.1 | -14.4 | 4.0 |

100 for reaching the final state and 0 for failures without partial credits. Following PerAct [3], we report the average success rates on 25 episodes for each task and the average success rates on all 18 tasks.

**Comparison with the State-of-the-art Methods.** We evaluate the Σ-agent five times on the same 25 episodes for each task and report the mean results due to the randomness of the sampling-based motion planner. The evaluation comprises two settings, training with 10 (Table 8) or 100 (Table 1) demonstrations per task. We re-implement the results of PolarNet (100 demos) and RVT (10 demos) since they are missing from the original paper. Other results are taken from the related literature. From Table 1 and 8, Σ-agent outperforms previous methods on both the 10 and 100 demonstrations by a large margin, up to $5.2\%$ and $5.9\%$ in average success rate over 18 tasks, respectively. In specific tasks, our Σ-agent achieves state-of-the-art performance on 13 out of 18 tasks for both the 10 and 100 demonstration settings.

Moreover, we compare Σ-agent with two state-of-the-art models, Act3D [12] and ChainedDiffuser [13], which are trained on $256 \times 256$ input resolution and tested on 100 episodes for each task. As shown in Table 2, our Σ-agent surpasses the two methods by $3.3\%$ and $2.3\%$ respectively, with **5x less training time**. Results of Σ-agent in other simulated environments are present in Appendix A.2.

**Contrastive Imitation Learning for Baselines.** To validate the effectiveness of contrastive IL, we integrate the contrastive IL module into other baseline models. We choose PolarNet [11] and RVT [1] as the baselines to demonstrate improvements in representations of point clouds and 3D re-rendered images. The main network and inference pipeline are kept unchanged, with only a contrastive IL module incorporated, adding a negligible increase in parameter count during the training process. As shown in Table 3, the contrastive IL module improves the performance of both PolarNet [11] and RVT [1] by $+2.8\%$ and $+1.8\%$ in average success rate over 18 tasks, respectively. Specifically, the performance of most tasks is improved (13 out of 18 and 11 out of 18), with a largest margin of 13.9% improvement. These improvements demonstrate two points: first, our proposed contrastive imitation learning can transfer across multiple models. Second, the learning method is effective for both 3D re-rendered images (RVT [1]) and point cloud representations (PolarNet [11]).

**Future State and Language Ablations.** We ablate the influence of future state and language contrastive IL as shown in Fig. 3 (a). We summarize three key points from the results. (1) Both language and future-state contrastive learning with current observations enhance performance. (2) Contrastive learning between current observations and language instructions accelerates the convergence speed of agent training, achieving higher performance at the early stage of training. (3) When future-state contrastive learning is added to a model that already includes language-state contrastive learning, the performance achieves a more substantial gain. We believe that the limited improvement in **only** utilizing future-state contrastive learning can be attributed to the insufficient demonstrations as the trajectory is the unit for constructing contrastive learning categories. Consequently, without leveraging language features, it is challenging to learn discriminative features.

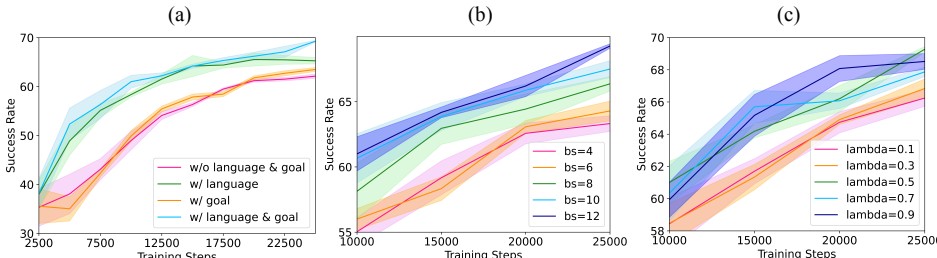

Figure 3: Ablation experiments. (a). The success rate of Σ-agent ablating language & future-state contrastive learning with current observations. (b). The success rate of Σ-agent ablating batch size of contrastive IL. (c). The success rate of Σ-agent ablating different $\lambda$.

**Batch-size Influence in Contrastive IL.** Contrastive training is sensitive to the scale of batch size [58, 53]. We vary the batch size as shown in Fig. 3 (b). It can be concluded that scaling up batch size plays a crucial role in boosting the agent's performance as it can include more negative samples.

**Coefficient $\lambda$ Ablations.** In Eqn. 4, $\lambda$ is the hyper-parameter that conditions the relation between contrastive IL and behavior cloning. We vary the $\lambda$ ranging from $[0, 1]$ to find the optimal value. From Fig. 3 (c), we train the agent with five different $\lambda$ values and observe it makes slightly different results. We choose $\lambda = 0.5$ for agent training as it is the relatively optimal value.

## 4.2 Real-world Experiments

We conduct experiments on a real robot, a 6-DoF UR5 robotic arm. The Σ-agent is validated on 5 real-world tasks, including a total of 9 variants. For each task, we collect 10 human demonstrations and train the Σ-agent with a single policy from scratch using all task demonstrations. Details of the real-world setup and data-collecting are provided in Appendix B. Table 4 presents the real-world results. We test the Σ-agent in 10 episodes for each task, and it achieves an average success rate of 62% across all tasks. We show the failure cases in Fig. 8.

Table 4: Performance of Σ-agent on 5 real-world tasks.

| Task | Succ. % |
|---|---|
| Stack cups | 60 |
| Put fruit in plate | 90 |
| Hang mug | 60 |
| Put item in barrel | 50 |
| Put tennis in mug | 50 |
| Average | 62 |

To analyze the reasons for failure: first, the limitation of a single front-view camera cannot provide precise visual information for tasks that require aiming, such as "Put tennis in barrel" and "Stack cups". Second, during the grasping process, imperfect grasp poses result in the translation or orientation of objects, worsening the collision problems. In the future, we plan to add an extra RGB-D camera at the wrist position to provide a first-person view. Additionally, integrating a pose estimation model for objects into Σ-agent would improve grasping poses and avoid collisions.

## 5 Conclusions and Limitations

In this work, we propose contrastive IL, a plug-and-play imitation learning strategy for language-guided multi-task 3D object manipulation. The contrastive IL optimizes the original imitation learning framework by integrating the contrastive IL module to refine both the feature extracting and interacting. Based on the contrastive IL, we design an end-to-end imitation learning agent Σ-agent utilizing the re-rendered virtual images from RGB-D input. Σ-agent is effective and efficient in both simulated environments and real-world experiments. However, we identify some limitations that exist. First, the scale and the number of camera views of our real-world experiments are limited. Second, the contrastive IL module in our method only demonstrates the effectiveness of closed-set scenarios like RLBench, but does not show the applicability in open-vocabulary scenarios. Lastly, like previous work, the generalization capacity of Σ-agent is limited, as shown in Table 7. We leave these issues for future work.

**Acknowledgments**

This work was supported by the Meituan Academy of Robotics Shenzhen and Guangzhou-HKUST(GZ) Joint Funding Program (Grant No.2023A03J0008), Education Bureau of Guangzhou Municipality.

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

| Task | Language Template | # of Variations | Avg. Keyframes |
|---|---|---|---|
| open drawer | "open the __ drawer" | 3 | 3.0 |
| slide block | "slide the __ block to target" | 4 | 4.7 |
| sweep to dustpan | "sweep dirt to the __ dustpan" | 2 | 4.6 |
| meat off grill | "take the __ off the grill" | 2 | 5.0 |
| turn tap | "turn __ tap" | 2 | 2.0 |
| put in drawer | "put the item in the __ drawer" | 3 | 12.0 |
| close jar | "close the __ jar" | 20 | 6.0 |
| drag stick | "use the stick to drag the cube onto the __ target" | 20 | 6.0 |
| stack blocks | "stack __ __ blocks" | 60 | 14.6 |
| screw bulb | "screw in the __ light bulb" | 20 | 7.0 |
| put in safe | "put the money away in the safe on the __ shelf" | 3 | 5.0 |
| place wine | "stack the wine bottle to the __ of the rack" | 3 | 5.0 |
| put in cupboard | "put the __ in the cupboard" | 9 | 5.0 |
| sort shape | "put the __ in the shape sorter" | 5 | 5.0 |
| push buttons | "push the __ button, [then the __ button]" | 50 | 3.8 |
| insert peg | "put the __ peg in the spoke" | 20 | 5.0 |
| stack cups | "stack the other cups on top of the __ cup" | 20 | 10.0 |
| place cups | "place __ cups on the cup holder" | 3 | 11.5 |

Table 5: **Tasks in RLBench.** We evaluate agents on 18 RLBench tasks, which include 249 variations like PerAct [3] does.

Table 6: Multi-task performance in the Ravens [25] environment. The results show the effectiveness of both Σ-agent and contrastive IL ($\mathcal{L}_{CL}$ in table).

| Methods | Put Blocks in Bowl | Pack Box Pairs | Packing Google Object Seq | Packing Google Objects Group | Stack Block Pyramid | Align Rope |
|---|---|---|---|---|---|---|
| CLIPort [27] | 92.3 | 88.8 | 80.1 | 85.4 | 75.0 | 51.2 |
| Σ-agent w/o $\mathcal{L}_{CL}$ | 95.5 | 94.1 | 81.3 | 87.2 | 79.1 | 60.8 |
| Σ-agent w/ $\mathcal{L}_{CL}$ | **96.7** | **97.3** | **82.1** | **87.6** | **83.5** | **65.3** |

## A   Simulation Experiments

### A.1   RLBench Tasks

The RLBench setting in our paper is elaborated in Table 5. We follow the PerAct [3] to use the 18 tasks with 249 variations in the RLBench. The examples of the 18 tasks and corresponding human instructions of specific variants in these tasks are shown in Fig. 4.

### A.2   Effectiveness in Other Simulated Environments.

Ravens [25] is a simulated benchmark environment built with PyBullet [59] for robotic rearrangement based on a Universal Robot UR5e. There are 3 simulated 640x480 RGB-D cameras covering the 0.5×1m tabletop workspace from the front, left shoulder and right shoulder. We train the model on six tasks in the Ravens environment [25] as shown in Table 6, selecting from the *seen* split of CLIPort [27]. The models are trained on 100 demonstrations and evaluated on 100 evaluation instances for each task. The Ravens benchmark scores 0 for task failure and 100% for success. The

Table 7: Evaluation on the generalization capacity of models. We test the models trained on 18 seen tasks directly on the 14 unseen tasks selected from the 74 tasks of RLBench [24].

| Method | 18 seen tasks | 14 unseen tasks |
|---|---|---|
| PerAct [3] | 49.4 | 1.52 |
| RVT [1] | 62.9 | 2.95 |
| Σ-agent | **68.8** | **3.81** |

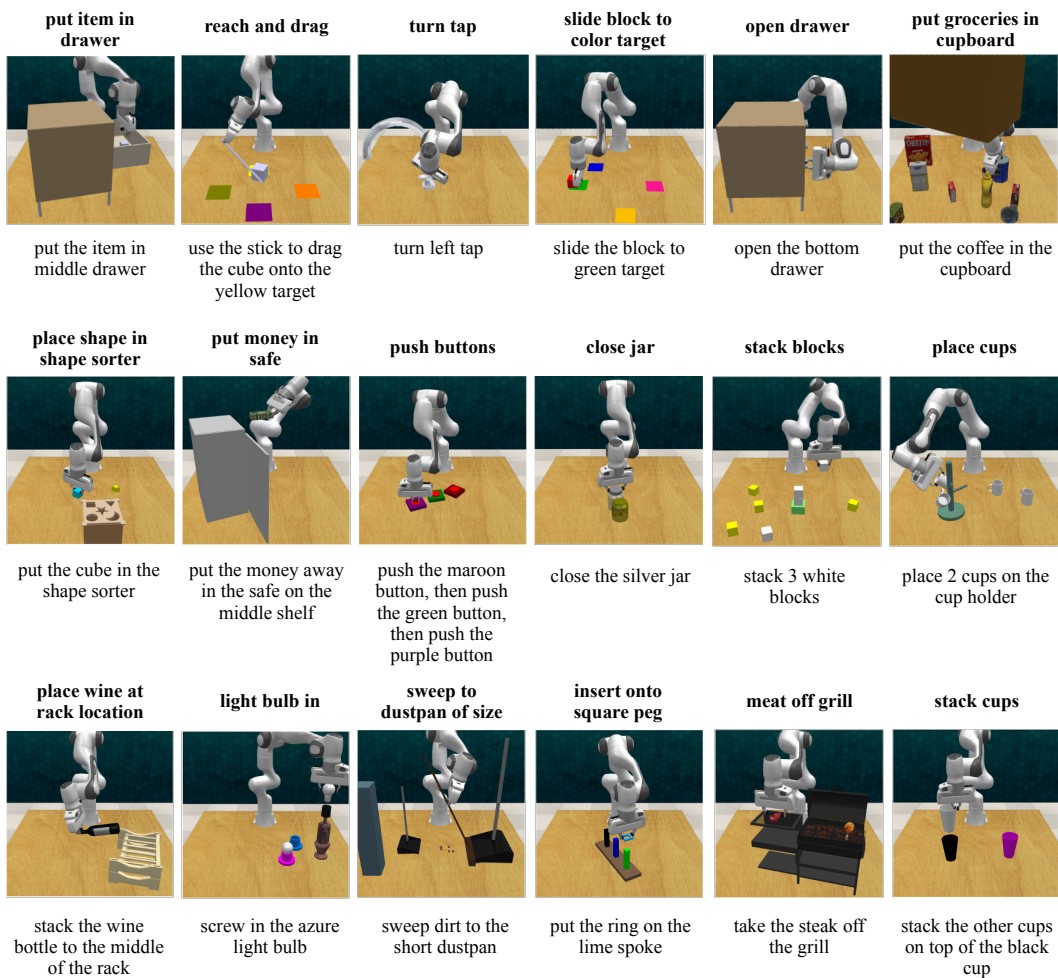

Figure 4: Examples of the 18 RLBench tasks (front view) with corresponding human instructions.

Table 8: Multi-Task performance rained with 10 episodes and evaluated using 25 episodes per task on RLBench (3 times average). The input resolution is $128 \times 128$. In `Train Time`, the $d$ represents days and $h$ represents hours. All other results are success rates, measured in percentage (%).

| | stack blocks | screw bulb | put in safe | place wine | put in cupboard | sort shape | push buttons | insert peg | stack cups | place cups |
|---|---|---|---|---|---|---|---|---|---|---|
| I-BC(CNN) [2] | 1.8 | - | 4.0 | 4.0 | 0 | 0 | 20.0 | 0 | 0 | 0 |
| I-BC(ViT) [2] | 4.4 | - | 16.0 | 8.0 | 8.0 | 0 | 24.0 | 0 | 0 | 0 |
| C2FARM [4] | 22.7 | - | 28.0 | 12.0 | 4.0 | 40.0 | 60.0 | 12.0 | 28.0 | 72.0 |
| PerAct [3] | 30.0 | ~16d | 68.0 | 32.0 | **72.0** | 68.0 | 72.0 | 16.0 | 32.0 | 36.0 |
| RVT [1] | 42.1 | ~23h | **77.3** | 52.0 | 65.3 | 68.0 | 93.3 | 45.3 | 34.7 | 100.0 |
| Σ-agent | **47.3** | ~22h | 74.6 | **56.0** | 46.7 | **80.0** | **94.7** | **90.7** | **64.0** | **100.0** |
| | stack blocks | screw bulb | put in safe | place wine | put in cupboard | sort shape | push buttons | insert peg | stack cups | place cups |
| I-BC(CNN) [2] | 0 | 0 | 0 | 0 | 0 | 0 | 4.0 | 0 | 0 | 0 |
| I-BC(ViT) [2] | 0 | 0 | 0 | 4.0 | 4.0 | 0 | 16.0 | 0 | 0 | 0 |
| C2FARM [4] | 4.0 | 12.0 | 0 | 36.0 | 4.0 | 8.0 | 88.0 | 0 | 0 | 0 |
| PerAct [3] | 12.0 | **28.0** | 16.0 | 20.0 | 0 | **16.0** | 56.0 | 4.0 | 0 | 0 |
| RVT [1] | **13.3** | 25.3 | 40.0 | 57.3 | 14.7 | 6.7 | 61.3 | 4.0 | 0 | 0 |
| Σ-agent | 5.3 | 14.6 | **45.3** | **62.6** | 18.6 | 14.7 | **72.0** | 5.3 | 4.0 | 1.3 |

evaluation also assigns partial credit for different tasks, such as $40\%$ for packing 2 out of 5 objects required by the instruction. The results in Table 6 show improvements in both our Σ-agent and contrastive IL, surpassing the CLIPort [27] by a large margin in all six tasks.

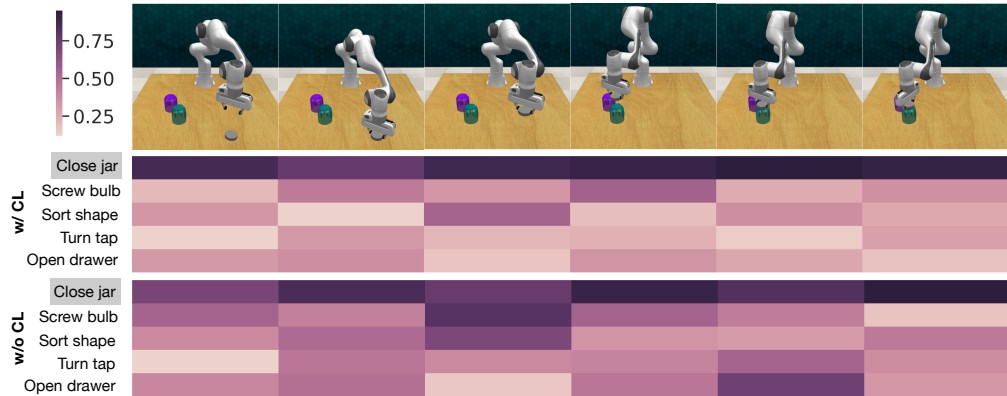

Figure 5: We visualize the similarity of a key-point trajectory of `close jar` task with multiple tasks' language instructions. Training with contrastive IL module maximizes the similarity between the visual observations and related language instructions (deeper color), reducing the similarity with negative instructions (lighter color).

### A.3 Visualization.

To qualitatively illustrate the effectiveness of contrastive IL, we visualize the multi-modal similarity between visual observations and language instructions. As shown in Fig. 5, we compute the cosine similarity between the embeddings of observations in a trajectory of `close jar` and different language instructions. For Σ-agent without contrastive IL module, we utilize the original text projection layer of CLIP [53] to project the textual embeddings for similarity computation. It is evident that contrastive IL amplifies the alignment between language instructions and local visual observations. For example, in the third keyframe, the agent without CL tends to confuse the `close jar` task with `screw bulb` and `sort shape`, as the three task instructions have close similarity with the visual observation.

## B    Real-world Experiments

### B.1    Hardware Setup

We conduct real-world experiments on a table-top setup with a 6-DoF UR5 robotic arm. The visual input is provided by a third-person perspective Orbbec Femto Bolt [2] (RGB-D) camera mounted directly above and facing forward. The camera streaming the RGB-D images of $1280 \times 960$ (hardware D2C align) at 30 Hz. The images will be resized to $640 \times 480$ for feeding into the model. The extrinsics of the RGB-D camera and robotic arm are calibrated via hand-eye calibration. We mount the ARUCO [3] AR marker in the base of UR5 arm. Additionally, we mount a DJI Osmo Action 4 [4] for recording inference demos. The hardware setup is shown in Fig. 6.

### B.2    Data Collection

We collect the human demonstrations by manual teaching, dragging the robotic arm to pre-defined keypoints and collecting the visual observations and gripper poses. These poses are executed with a motion-planner using ROS [5] and MoveIt [6]. We define 5 tasks to experiment, including `Stack cups`, `Put fruit in plate`, `Hang mug`, `Put item in barrel`, and `Put tennis in mug`. We

---

[2]https://www.orbbec.com/products/tof-camera/femto-bolt/
[3]https://github.com/pal-robotics/aruco_ros
[4]https://www.dji.com/cn/osmo-action-4
[5]https://ros.org/
[6]https://docs.ros.org/en/kinetic/api/moveit_tutorials/html/

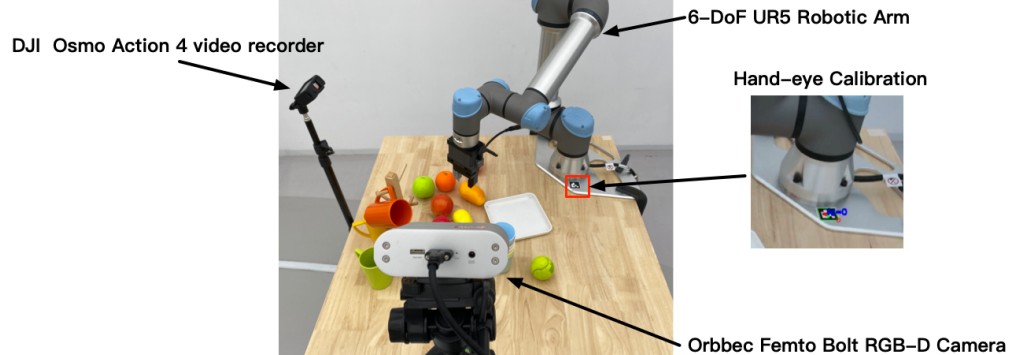

Figure 6: Real robot setup with Orbbec Femto Bolt and UR5.

| Task | Language Template | # of Variations | Variants Category |
|------|-------------------|-----------------|-------------------|
| stack cups | "stack the __ cup on top of the __ cup" | 2 | color |
| put fruit in plate | "put the __ in the plate" | 2 | object |
| hang mug | "hang the __ mug on the rack" | 2 | color |
| put item in barrel | "put the __ in the barrel" | 1 | object |
| put tennis in mug | "put the tennis in the __ mug" | 2 | color |

Table 9: **Tasks in real-world.** There are 5 tasks and 9 variants totally.

collect 10 demonstrations for each task. The details of the collected data samples are shown in Table 9 and Fig 7.

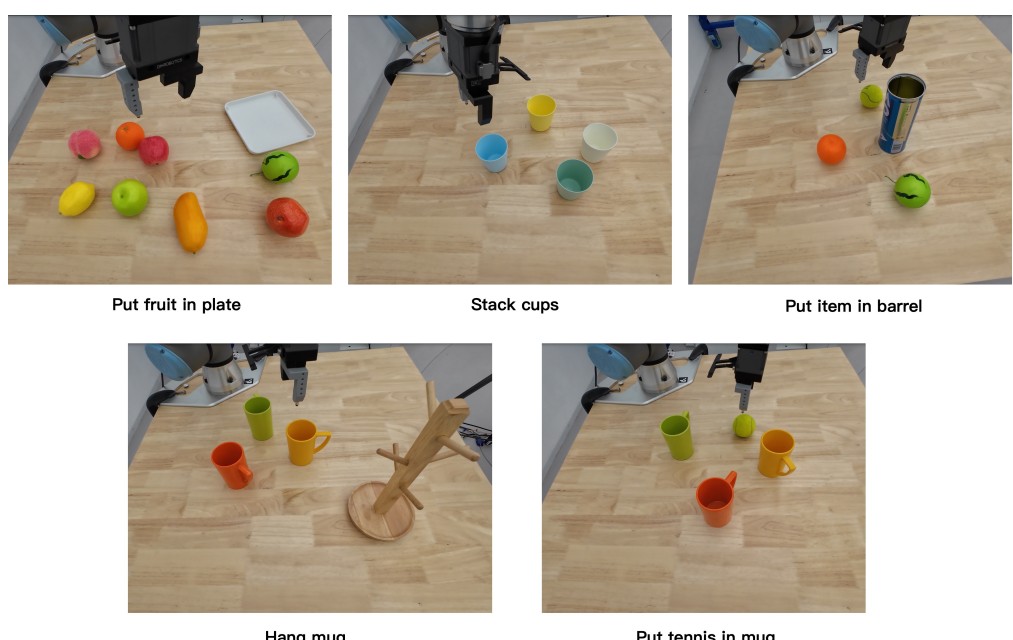

Figure 7: Illustration of the 5 real-world tasks.

## B.3 Training and Evaluating

The Σ-agent is trained on 50 demonstrations for 5 tasks. The training samples are augmented with $\pm 0.125m$ translation perturbations and $\pm 45°$ yaw rotation perturbations following PerAct [3]. The

training is from scratch, not fine-tuning based on checkpoints trained in simulated environments. For evaluation, Σ-agent is validated on 10 episodes for each task.

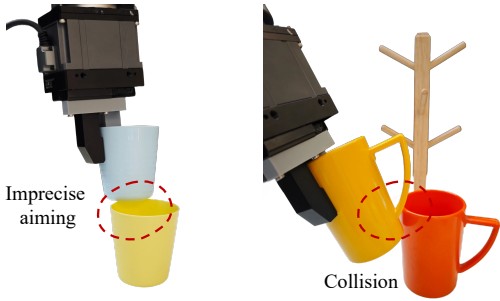

Figure 8: Failure cases.

## C Additional Model Details

### C.1 Action Prediction

Based on the context features $\mathbf{v}$ after feature fusion, the decoder outputs the 6-DoF end-effector pose (3-DoF for translation and 3-DoF for rotation), the gripper state (open or close) and a binary value for whether to allow collision for the motion planner. We simply utilize a 2D convolution layer and bi-linear upsampling to decode and upsample the encoded context features to the original rendered image size ($220 \times 220$). Following RVT [1], Σ-agent predicts the heatmaps for the 5 virtual views, and the heatmaps will be projected back to the 3D space to predict the point-wise scores for the robot workspace. Then, the translation of the end-effector is determined by the 3D point with the highest score. The rotations, gripper state and collision indicator are predicted based on the max-pooled image features and the sum of image features weighted by the heatmaps. Suppose the $\mathbf{h}$ is the view-wise heatmaps predicted, the features for predicting rotations, gripper state and collision indicator are formulated as:

$$f = [\texttt{sum}(\mathbf{v} \odot \mathbf{h}), \texttt{maxpool}(\mathbf{v})] \tag{5}$$

where $\texttt{sum}$ and $\texttt{maxpool}$ denote the sum and max-pooling operation across the spatial dimension of tokens. $\odot$ represents the element-wise multiplication between the context features and heatmaps.

Then, following the PerAct [3] and RVT [1], we utilize the Euler angles representation for the rotation, and each angle is discretized into bins of $5°$ for $dx, dy, dz$. In that case, the rotation prediction is converted to a classification problem, where the agent is trained to classify the angles into 216 categories ($3 \times 360°/5°$). Hence, we project the features $f$ onto a 220-dimensional space using a linear layer. Within this space, 216 dimensions are allocated for rotation prediction, while 2 dimensions each are dedicated to binary gripper state prediction and binary collision state prediction.

For the training loss for action prediction, we use the cross-entropy loss for the translation and rotation. Binary classification loss is utilized for the gripper state and collision state. Contrastive loss is utilized besides the aforementioned losses to supervise representation learning in the contrastive IL module.

### C.2 Q-Function Analysis

Same as PerAct [3], we decode the features to estimate the Q-function of action-values, as $\mathcal{Q}(a_t|s_t, l)$. The $\mathcal{Q}(a_t|s_t, l)$ is equivalent to the transition probability $\mathcal{P}_\pi$ of state $s_t$ and $l$ to the next state $s_{t+1}$ under the discounted state occupancy measure [19, 17, 20]:

$$\mathcal{Q}_\pi(a_t|s_t, l) \triangleq \mathcal{P}_\pi(s_{t+1}|s_t, l) \tag{6}$$

When the $g^+$ is the next state of $s_t$ in Eqn. 3, the $f_{\delta,\varphi}$ maximizes the similarity between $(s_t, l)$ and $s_{t+1}$. In other words, training $f_{\delta,\varphi}$ aims to maximize the transition probability from $s_t$ to $s_{t+1}$ with the language instruction $l$ by minimizing the distance between corresponding pairs $[(s_t, l), s_{t+1}]$, while simultaneously maximizing the distance of negative pairs. Therefore, training of $f_{\delta,\varphi}$ with $\mathcal{L}_{(s_t, l) \leftrightarrow g = s_{t+1}}$ is beneficial for maximizing the $\mathcal{P}_\pi(s_{t+1}|s_t, l)$, thereby maximizing the $\mathcal{Q}_\pi(a_t|s_t, l)$. $f_{\delta,\varphi}$ can be an extra critic function to facilitate the policy $\pi$ mimicking target policy $\pi^+$ in the level of representation learning.

## D   Related Multi-Task Learning in RL

Learning a single agent for multiple tasks is of vital significance to robotic learning. One of the main challenges in multi-task learning is the conflicting representations and gradients among different tasks. Previous works address multi-task learning via strategies like knowledge transfer [60, 61], representation sharing [62, 63, 64], and gradient surgery [65]. With the advent of large vision-language models [53, 21, 22, 66] and LLMs [67, 68, 69], language instructions serve as complementary hints to differentiate task representations in policy learning [31, 32, 16, 11, 3, 1, 12, 13, 14]. In this paper, we leverage contrastive learning to amplify the distinction function of linguistic hints.

