# OpenReview forum: "Contrastive Imitation Learning for Language-guided Multi-Task Robotic Manipulation"
_robot-learning.org/CoRL/2024/Conference — CoRL 2024_

### Official Review · Reviewer_Ejqe · 2024-07-21
**Well written paper. Some concerns about novelty and implementation**

**Originality:** 2
**Technical Quality:** 3
**Clarity Of Presentation:** 4
**Potential Impact:** 2
**Recommendation:** 2
**Confidence:** 4

**Review:**

The paper is generally well written and easy to follow. The introduction lays out the problem motivation well, the related works section is thorough, and the method section is well laid out. The experiments and comparisons are also thorough, with the ablations being informative in breaking down the effect of the language vs. goal based contrastive learning. However I have a few specific concerns:
* The work, for the most part, has somewhat low technical novelty. Adding auxiliary contrastive learning objectives is a useful tool that has been established in numerous works, as the authors point in the related works section. And while there is an improvement in performance compared to prior work like RVT, the performance gain of ~5% is somewhat small.
* line 187 states: “for the goal and state-language contrastive training, we utilize the next state in the trajectory as the goal state.” Why is only the immediate next step chosen as the goal state? It seems like a rather constraining choice. I would be curious to see what happens if set the goal state as the state 10 steps ahead, or the final state of the trajectory. Additionally, the ablation in Fig 3a does not really show any gains from incorporating the goal-based contrastive objective. This makes sense if the goal state we are choosing is only a single tilmestep away from the current state.
* I’d like to see a comparison to RVT2, which claims to provide state-of-the-art results with significant gains over RVT. The contrastive learning techniques used in this paper can be applied to RVT2, and it would be interesting to see whether we there is an improvement in that setting as well.
* The limitations mentioned in the final section are rather generic and not specific to the ideas presented in this paper. I would have liked to see the authors point out specific observations where the method is not performing as well as it should, and propose some ideas to address these limitations.

**Quality Of The Limitations Section:**

2

**Questions For Rebuttal:**

Please see the questions I raised in my review above.

**Robotics Focus:**

4

**Summary Of Paper:**

The authors introduce an imitation learning method that incorporates contrastive learning in the training objective to better align the underlying vision and language representations and improve downstream policy performance. The contrastive learning objective is two fold: (1) aligning states and language representations for the same task, and (2) aligning joint state-language and goal representations for the same task. The authors also propose a Multi-View Querying Transformer to improve the training efficiency of the model. On the simulated RLBench suite, the authors show that the model generally outperforms existing approaches, including RVT.

**Summary Of Recommendation:**

I think at least one of the following is needed: (1) more technical novelty; (2) larger performance gains over the current state-of-the-art methods.

---

### Official Review · Reviewer_HHmD · 2024-07-25
**Review of Contrastive Imitation Learning for Language-guided Multi-Task Robotic Manipulation**

**Originality:** 3
**Technical Quality:** 3
**Clarity Of Presentation:** 4
**Potential Impact:** 3
**Recommendation:** 2
**Confidence:** 4

**Review:**

Strengths:
- This work presents a contrastive learning-based approach to learning from both vision and language modalities for robotics. To the best of my knowledge, this is novel to the robotics community and is also an important problem.
- The experiments show that the contrastive representations enable some performance improvements compared to the non-contrastive representations.
- This method can be applied to real-world problems with relatively low amounts of demonstration data. The additional cost to training is also low.
- Writing is clear and easy to follow.

Weaknesses:
- The improvements are minor (2-3% improvement in success rate), especially compared to differences between different base models (e.g., between PolarNet and RVT), which suggests that the impact of the proposed contrastive representations is less significant compared to other model choices. Perhaps, for this particular setting with relatively few tasks, aligning vision and language modalities is not as important.
- Given that there is prior work on aligning text and vision modalities with contrastive losses [1, 2], the main difference in this work is additionally contrasting the <image, language> to goal image representations. This additional loss, however, has less impact on performance according to the ablations in Figure 3(a).
- The related work section could be improved by including references to works on (contrastive) alignment of vision-language modalities outside of robotics.

[1] Jia et al. Scaling Up Visual and Vision-Language Representation Learning With Noisy Text Supervision. ICML 2021.
[2] Li et al. Align before Fuse: Vision and Language Representation Learning with Momentum Distillation. NeurIPS 2021.

**Quality Of The Limitations Section:**

3

**Questions For Rebuttal:**

Clarification questions:
- In Table 2, is $\Sigma$-agent using RVT as the base model? If so, what is the performance of RVT in the 100-episode per task setting?
- How much training time does contrastive learning add to RVT and to PolarNet? In Table 1, is it correct that $\Sigma$-agent applies the contrastive learning module to RVT and is faster to train compared to RVT?
- For real robot experiments, how does the performance compare to the base model without the contrastive representation?

Other questions:
- What would happen if this is to scaled to a much larger number of tasks? How does dataset size and number of training tasks affect performance?
- Does this alignment of vision and language modalities enable better generalization to new tasks and task variations?
- Currently, the variations in tasks are reference variations in the color and object instance. Have the authors tried testing other variations such as position/direction (e.g., "stack the cup located in the top-right corner on top of the cup in the bottom-left corner"), size (e.g. "stack the smaller cup on top of the bigger cup"), and shape? It would also be interesting to organize the results in terms of different types of variations to see which variations are harder for the model to handle.
- Have the authors tried varying the camera pose? Some of the aforementioned variations (position/direction) depend heavily on the camera perspective. It'd be interesting to see whether aligning the vision-language modalities allow generalization to new camera poses and references to position/direction.

**Robotics Focus:**

4

**Summary Of Paper:**

This paper introduces an imitation learning approach that leverages contrastive learning to align the image + language embeddings and the <image, language> + goal image embeddings. The vision encoder is additionally trained on re-rendered virtual images (following RVT). The evaluations are conducted on RLBench and a set of real-world manipulation tasks, and demonstrates improvements over RVT in the multi-task setting.

**Summary Of Recommendation:**

This paper studies an important problem that is relevant to the robot learning community, and presents an approach that is simple to integrate into existing approaches. However, from the present experiments, it's not entirely clear to me that there are strong advantages, and stronger analysis may highlight where this approach would be most beneficial.

---

### Official Review · Reviewer_wN6C · 2024-07-25
**Review 1**

**Originality:** 2
**Technical Quality:** 3
**Clarity Of Presentation:** 3
**Potential Impact:** 2
**Recommendation:** 3
**Confidence:** 4

**Review:**

Quality: The paper quality is quite high: the content is well organized under appropriate subsections, and the experiments and results are presented elegantly. There are few, if any, spelling or grammatical errors.

Clarity: Some sections of the paper are clear (e.g., the experiments section onwards is clear), while others have room for improvement in terms of clarity and rigor. For example, the introduction does not clearly articulate the core research problem (and its importance) addressed by the paper (i.e., lines 34-35). The essence of the method is actually quite simple, yet Figure 2 overcomplicates it – and the caption of Figure 2 contains symbols that are not represented in the figure, which is a point of confusion. There are terms used to describe the methodology that are not defined beforehand, e.g., what is a “discriminative feature” and what does “interacting feature” mean? I recommend defining these terms before use.

Originality: The idea of using auxiliary contrastive learning objectives to promote or bolster policy learning in an imitation learning setup is not original. However, the specific objectives used here appear to be original.

Significance: Objectives to regularize various aspects of robotic policy learning are relevant to the robot learning community. However, the contributions proposed in this paper yield marginal improvements over existing baselines.

Summary of strengths:
- The proposed method is easy to integrate with existing imitation learning pipelines and is perhaps a cheap solution to slightly boost policy performance.
- The evaluation of the method and baselines is quite comprehensive and well presented. The reviewer appreciates the additional ablations conducted both on the proposed method (Figure 3) and the baselines (Table 3). The analysis is also quite clear.
- The proposed contrastive learning objectives are shown to reduce training times.

Summary of weaknesses:
- The results are a bit underwhelming. While the proposed contribution performs best in aggregate by ~6%, it is unclear why the contrastive objectives harm performance on some tasks. Similarly, the contrastive learning objectives do not yield significant increase in success rates when implemented on the baselines (<3% in aggregate).
- The ablation in Figure 3 indicates that one of the two proposed objectives (goal contrastive loss) contributes minimally to policy performance.
- The limitations stated in Section 5 are more so general limitations of language-conditioned imitation learning rather than concrete limitations associated with the proposed method.

**Quality Of The Limitations Section:**

2

**Questions For Rebuttal:**

- A more compelling and intuitive motivation should be established for each of the contrastive learning terms. At the present moment, the motivation for these terms in the form of a need for feature “alignment” is vague to me. The language-state term is more intuitive than the goal term.
- Can the authors please explain the discrepancy of results across tasks? Why does contrastive learning benefit some but harm others? Can this be improved?
- Quality / Clarity: I encourage the authors to simplify Figure 2 and address the above comments pertaining to quality and clarity.

**Robotics Focus:**

4

**Summary Of Paper:**

The paper titled “Contrastive Imitation Learning for Language-guided Multi-Task Robotic Manipulation” presents a language-conditioned imitation learning framework for robotic manipulation. The contribution consists of augmenting the standard imitation learning objective with two auxiliary contrastive learning objectives to promote alignment between 1) the language instruction and the visual representation and 2) the language instruction + visual representation and the goal. This objective is used to train a multi-task, multi-view policy architecture, which is evaluated against a comprehensive set of baselines. The simulated results indicate that 1) the overall proposed system (new objective + policy architecture) outperforms existing methods and 2) contrastive objectives marginally improve the performance of baselines. The proposed method is validated on a real robot.

**Summary Of Recommendation:**

The paper is in relatively good condition in that it is well organized, quite clear, and experiments are extensive. The contribution contains some originality, and while it does improve policy performance (in aggregate) and learning efficiency, the significance of the results (and hence the contribution) is questionable. On these bases, my recommendation is weak reject.

---

### Official Review · Reviewer_h5Zm · 2024-07-26
**CoRL 2024 Conference Submission269 Reviewer h5Zm**

**Originality:** 3
**Technical Quality:** 3
**Clarity Of Presentation:** 4
**Potential Impact:** 3
**Recommendation:** 3
**Confidence:** 4

**Review:**

*Strengths*:
- Contrastive imitation learning is skillfully designed to enhance the representation of language-conditioned manipulation tasks. It effectively utilizes discriminative features to accurately perceive the current state and align representations with task correlations.
- The simulation experiments are thorough, encompassing multiple manipulation tasks to demonstrate significant improvements over previous baselines. The efficiency of the proposed contrastive imitation learning modules is clearly showcased.
- The paper effectively formulates key questions for language-conditioned multi-tasking. It provides a comprehensive introduction to the proposed objectives and their detailed explanations.

*Weakness*:
- While the paper claims that the proposed MVQ-Former efficiently reduces the number of visual tokens, it would benefit from more in-depth analysis and concrete comparisons to substantiate this claim.
- In real-world experiments, the results for several simple pick-and-place tasks show unsatisfactory success rates. The reasons provided for these appear to be relatively straightforward to address, suggesting room for improvement in real-world experiments.
- The results of the simulated experiments indicate a significant improvement in tasks such as "put in drawer" and "close jar." However, the paper would benefit from a more detailed analysis explaining why the proposed method excels in these specific tasks.
- The baselines are reimplemented using different numbers of demonstrations for training. However, more detailed comparisons and analyses regarding the impact of varying demonstration numbers and the reasons why using different numbers of demonstrations would strengthen the paper’s claims.

**Quality Of The Limitations Section:**

3

**Questions For Rebuttal:**

- Efficiency of MVQ-Former: Can you please provide a more detailed analysis and concrete comparisons to substantiate the claim that the proposed MVQ-Former efficiently reduces the number of visual tokens?
- Real-World Experiment Improvement: The success rates in real-world pick-and-place tasks are not very satisfying. Could these rates be improved by relatively simple means, such as using multi-view cameras to provide more precise visual information or adding an extra RGB-D camera at the wrist position as proposed? If these improvements are feasible, please provide several results.
- Task-Specific Improvements: In simulated experiments, the method shows a significant improvement on tasks like "put in drawer" and "close jar." Can you please provide a more detailed analysis explaining why the proposed method excels in these specific tasks?
- Baseline Reimplementation and Demonstration Variability: The baselines are reimplemented using different numbers of demonstrations for training. Can you include more detailed comparisons and analyses on the impact of varying demonstration numbers on the performance of the proposed method?

**Robotics Focus:**

4

**Summary Of Paper:**

This paper introduces an end-to-end imitation learning framework for language-conditioned multiple 6-DoF manipulation tasks for agents. It presents a novel method called contrastive imitation learning to refine the visual representation and also the vision-language representation. It also proposes the Multi-View Querying Transformer to improve efficiency by minimizing the token numbers. Experiments are performed on several benchmarks and various manipulation tasks. The contrastive IL module is integrated with baseline models to demonstrate its effectiveness in different visual representations.

**Summary Of Recommendation:**

Overall it is a good work with comprehensive results in the simulated experiments. The paper is easy to read and the proposed method is easy to understand. I will be more confident in recommending the paper for acceptance if the questions are well answered or addressed.

---

### Official Review · Reviewer_H222 · 2024-07-29
**Review of Submission269**

**Originality:** 3
**Technical Quality:** 3
**Clarity Of Presentation:** 4
**Potential Impact:** 2
**Recommendation:** 3
**Confidence:** 5

**Review:**

## Strengths

1. The paper presents an intuitive approach to incorporate contrastive learning into imitation learning for improving multi-task robotic manipulation.
2. The authors provide a comprehensive baselines in the RLBench experiments and also provide ablation studies analyzing the impact of different components of their method.
3. The paper is overall well-written, well-organized and easy to follow - great job on the writing!

## Weaknesses

1. While the addition of contrastive loss shows improvements, I am not convinced that they are significant. The paper does not report the variance of success rate across tasks in RLBench (which I think is important, given that RLBench uses a sampling-based planner AFAIK). Looking at the results in RVT [1] paper, it seems like the variance on some tasks can be as high as 7%. Considering this, the reported improvements (5.2% and 5.9% over RVT in 10 and 100 demo cases) may not be as significant.
2. I am not entirely sure what the purpose of some of the experiments and reported results in the paper is. For example, Table 2 compares, besides the success rate, the training time of proposed approach with Act3D and ChainedDiffuser showing that the proposed approach is significant faster to train, but fails to acknowledge here that RVT is probably equally faster. Also, I am not sure what the purpose of real-world experiments is if there are no baselines in it to compare with. RVT already demonstrates that you could deploy this pipeline on real system, and as authors acknowledge in the paper, during inference the proposed approach is no different from RVT.
3. While authors demonstrate that using the contrastive IL module with other approaches (say RVT) could lead to better results (a 1.8% improvement for RVT), it seems like the module seems to make RVT performance worse in majority of the tasks (13/18 tasks are worse with the module). I am not sure if the slight average success rate improvement is enough to claim that the contrastive learning module could potentially be useful in other architectures.

## Minor Comments/Nits

- The paper would benefit from a more detailed explanation of the MVQ-Former and how it’s different from the multi-view transformer in RVT paper, as this seems to be an important component of the architecture.
- Figure 2 is dense and a little harder to follow.
- Some parts should be more clearly defined with in the paper, e.g., I shouldn’t have to go through RVT paper to understand what collision state is in the action space.

[1] RVT: Robotic View Transformer for 3D Object Manipulation

**Quality Of The Limitations Section:**

2

**Questions For Rebuttal:**

In addition to the questions in the previous sections, I have the following questions:

1. Do you have any qualitative examples of why the proposed approaches performs better than say RVT in some of the RLBench tasks?

2. In limitations, you state that the large discrepancy between RLBench and real-world environments leads to failure of sim-to-real transfer. Could you clarify this statement? Do you mean that “conclusions made in sim possibly may not hold in real-world” or “training in sim and deploying zero-shot in real-world wouldn’t work”? I usually associate sim-to-real with the latter, but that doesn’t make a lot of sense in this paper’s setting.

**Robotics Focus:**

4

**Summary Of Paper:**

This paper introduces a contrastive imitation learning approach for language-guided multi-task robotic manipulation. The paper builds upon the RVT [1] with the key additions being (i) the integration of contrastive learning into the imitation learning framework to refine vision feature extraction and vision-language feature interaction, and (ii) introducing a multi-view querying transformer for efficient contrastive RL. The authors evaluate their method on the RLBench benchmark demonstrating some improvements over state-of-the-art methods. The authors also test their method on a few real-world tasks.   [1] RVT: Robotic View Transformer for 3D Object Manipulation

**Summary Of Recommendation:**

The main contribution of the paper is the idea of using contrastive learning in the RVT framework, and I think the results from RLBench don't necessarily show yet that this led to significant improvement in performance.

---

### Official Review · Reviewer_6fNC · 2024-07-29
**Official Reivew - Submission 269**

**Originality:** 3
**Technical Quality:** 4
**Clarity Of Presentation:** 4
**Potential Impact:** 3
**Recommendation:** 3
**Confidence:** 4

**Review:**

### Strengths

- The proposed contrastive imitation learning modules and MVQ-Former seem to be novel enough in the context of multi-task learning and $\Sigma$-agent outperforms previous state-of-the-art methods in RLBench and Ravens.
- The proposed contrastive IL module is easy to plug into previous methods such as PolarNet and RVT to improve its performance on RLBench.
- The experiments show how Σ-agent can be deployed on a real robot with a 62% success rate across 5 different tasks. Also, the authors include nice videos of some real robot executions.
- The paper is well written and the visualization in Appendix Figure 5 is very nice to show that the learned visual representations are well aligned with the task instructions representations.

### Weaknesses
- The contrastive IL module improvements when integrated with PolarNet and RVT seem marginal, as it only slightly improves its original performance by +2.8 and +1.8 respectively.
- The real-world experimental videos are missing the failure cases. It would be nice to see some of the failure modes to understand better the method limitations.
- The performance in the ablation graph Figure 3 (a) w/ language and goal and w/ language seems very close and especially in the last data point seems that adding a goal is very beneficial. What happens when the training is longer than 25K steps? Have the performances already plateaued by this point?
- It is a bit strange that Table 2 comes before Table 1 in the text.

**Quality Of The Limitations Section:**

3

**Questions For Rebuttal:**

Please, take a look at the weaknesses in the review. Questions should be addressed to make the paper stronger.

**Robotics Focus:**

4

**Summary Of Paper:**

This paper proposes $\Sigma$-agent, a novel multi-task robotic manipulation learning agent. $\Sigma$-agent introduces new contrastive imitation learning modules that help to learn discriminative features to accurately perceive the current state and also to align task and language representations better. Then, a multi-view querying transformer module (MVQ-Former) is proposed to aggregate visual information into a set of 5 learnable queries (one per available input camera) that are fused later on with language features to produce the context features from which the final robot action is decoded. $\Sigma$-agent outperforms other baselines on 18 RLBench tasks. It shows how the proposed contrastive imitation learning modules can be plugged during training to previous methods (RVT and PolarNet) to improve its final performance. Finally, the paper demonstrates that the method can be deployed on a real-robot performing 5 different manipulation tasks.

**Summary Of Recommendation:**

My recommendation is Weak Accept. I believe that $\Sigma$-agent and specifically the proposed contrastive imitation learning modules will benefit the robotics community. While contrastive learning already exists and has been well studied, the proposed modules are novel and can be easily integrated with other language-guided robotic manipulation agents. However, when integrated with RVT and PolarNet, the performance only improves slightly. Further discussion with the authors and other reviewers' points of view can help towards the final decision.                                                             ||||||    Update: I want to thank the authors for the effort and work put into the rebuttal to respond to the 7 reviews. I will keep my final recommendation as Weak Accept. The authors addressed most of my concerns. However, I still believe that the improvements of this work when plugged into other methods remain very limited.

---

### Official Review · Reviewer_9vfD · 2024-07-31
**summary of Contrastive Imitation Learning**

**Originality:** 4
**Technical Quality:** 4
**Clarity Of Presentation:** 3
**Potential Impact:** 3
**Recommendation:** 3
**Confidence:** 3

**Review:**

**Summary**
This study presents Σ-agent, an end-to-end imitation learning agent designed to address the challenge of language-guided multi-task robotic manipulation. The core innovation lies in the application of Contrastive Imitation Learning (CIL) to enhance the understanding of vision-language and current-future representations. The Σ-agent introduces a Multi-View Querying Transformer (MVQ-Former) to aggregate semantic information and demonstrates significant performance improvements across 18 RLBench tasks, surpassing state-of-the-art methods.

In the methodology section, the article details the design of the Σ-agent, particularly how it utilizes contrastive learning to optimize feature extraction and vision-language interaction. Experimental results indicate that the Σ-agent achieves average success rates of 5.2% and 5.9% improvement in 10 and 100 demonstration training, respectively. Additionally, it achieves a 62% success rate in five real-world manipulation tasks, demonstrating its effectiveness in real-world scenarios.

**Strengthens**

1.This paper innovatively integrates contrastive imitation learning into the imitation learning framework, enabling robots to better understand and differentiate multi-task language instructions.

2.The proposed MVQ-Former effectively aggregates information from multiple views, reducing computational complexity while enhancing model performance.

3.The experiments in both RLBench and real-world tasks demonstrates the effectiveness of the proposed method, showcasing its potential applications in multi-task learning and real-world environments.

**Weakness**

1.The model may still face challenges when handling complex tasks, particularly those requiring precise visual information, as the current visual input may not suffice.

2.Dependence on Human Demonstrations: What are the implications of relying on human demonstrations for training? How might this affect scalability and applicability in diverse environments?

3.Generalization to Unseen Tasks: How does the Σ-agent perform when faced with tasks that were not part of the training set? Is there evidence to suggest that the model can generalize effectively to novel tasks?

4.Future Directions: What are the next steps for improving the Σ-agent? Are there plans to enhance its robustness or expand its capabilities beyond the current scope?

**Quality Of The Limitations Section:**

2

**Questions For Rebuttal:**

See weakness.

**Robotics Focus:**

4

**Summary Of Paper:**

This study presents Σ-agent, an end-to-end imitation learning agent designed to address language-guided multi-task robotic manipulation.

**Summary Of Recommendation:**

Overall, this article proposes a novel learning approach in the field of robotic manipulation, effectively improving model performance through contrastive imitation learning, while still facing challenges in real-world applications.

---

### Decision · Program_Chairs · 2024-09-04

**Decision:**

Accept

**Comment:**

This paper proposes an end-to-end imitation learning method which introduces auxiliary contrastive learning objectives to align visual representations with language instructions as well as the language-visual representation with the goal. Additionally, a multi-view querying transformer is proposed as an important component for efficiently incorporating contrastive imitation learning into the RVT framework.

Overall, reviewers appreciated the motivation of the proposed method, the diverse simulated benchmark and real-world experiments, and the high writing quality. However, reviewers also raised significant questions and concerns. Of particular note:

- Most reviewers note that the empirical gains are marginal. Furthermore, the ablations suggest that the performance improvements can not necessarily be attributed to the novel contributions of the work.
- Numerous reviewers request clarity on comparing the work against RVT and RVT2, both in the form of discussions as well as analysis.
- Some reviewers raise questions about the novelty of the work, since prior robot learning works have also incorporated auxiliary contrastive losses.
- Some reviewers note that it’s unclear whether the method can scale to more complex or unseen tasks.

The reviewers raise other questions and also provide helpful suggestions for improvement. It would be helpful for authors to address these during the rebuttal phase.

Update: The authors provided thorough and helpful responses to the numerous concerns raised by 7 reviewers. After discussions in the post-rebuttal phase, most reviewers are positive about the work and appreciate the statistically significant (if marginal) performance improvements over a variety of strong methods. *This work will be a valuable contribution to CoRL, so I am recommending an acceptance*. However, I recommend that the authors address the remaining reviewer feedback, especially those regarding performance and generalization. One reviewer provides the following helpful suggestions for the camera-ready:
- calling this goal-state contrastive learning is somewhat misleading. one would not normally associate the next state as a "goal". the semantics here need to be rephrased and we need to see an updated manuscript with a better interpretation of what the role of this "goal-state" contrastive learning is.
- we need to better understand why both language and goal contrastive learning are needed, and that why without language contrastive learning we do not see many benefits to goal contrastive learning.